# Epibrassinolide Regulates *Lhcb5* Expression Though the Transcription Factor of MYBR17 in Maize

**DOI:** 10.3390/biom15010094

**Published:** 2025-01-09

**Authors:** Hui Li, Xuewu He, Huayang Lv, Hongyu Zhang, Fuhai Peng, Jun Song, Wenjuan Liu, Junjie Zhang

**Affiliations:** 1Institute of Quality Standard and Testing Technology Research, Sichuan Academy of Agricultural Sciences, Chengdu 611130, China; lihui8627@outlook.com (H.L.); drsjn@scsaas.cn (J.S.); 2College of Life Science, Sichuan Agricultural University, Ya’an 625000, China; hxiaowu@stu.sicau.edu.cn (X.H.); lhyal2o3@163.com (H.L.); 19180173072@163.com (H.Z.); p1525937628@163.com (F.P.)

**Keywords:** photosynthesis, antenna proteins, MYBR17, *Lhcb5*

## Abstract

Photosynthesis, which is the foundation of crop growth and development, is accompanied by complex transcriptional regulatory mechanisms. Research has established that brassinosteroids (BRs) play a role in regulating plant photosynthesis, with the majority of research focusing on the physiological level and regulation of rate-limiting enzymes in the dark reactions of photosynthesis. However, studies on their effects on maize photosynthesis, specifically on light-harvesting antenna proteins, have yet to be conducted. The peripheral light-harvesting antenna protein *Lhcb5* is crucial for capturing and dissipating light energy. Herein, by analyzing the transcriptomic data of maize seedling leaves treated with 24-epibrassinolide (EBR) and verifying them using qPCR experiments, we found that the MYBR17 transcription factor may regulate the expression of the photosynthetic light-harvesting antenna protein gene. Further experiments using protoplast transient expression and yeast one-hybrid tests showed that the maize transcription factor MYBR17 responds to EBR signals and binds to the promoter of the light-harvesting antenna protein *Lhcb5*, thereby upregulating its expression. These results were validated using an Arabidopsis *mybr17* mutant. Our results offer a theoretical foundation for the application of BRs to enhance the photosynthetic efficiency of maize.

## 1. Introduction

Brassinosteroids (BRs) are a class of steroidal compounds [1,2], of which 24-epibrassinolide (24-EBL) is the most active form [3]. BR has been shown to upregulate the expression of key genes of the photosynthesis pathway to improve the photosynthetic efficiency of maize. The treatment of rice with 24-Epibrassinolide (24-EBL) after heading has been found to increase both the chlorophyll content and the net photosynthetic rate [4]. Treating geranium, maize, and pea leaves with EBR can increase the chlorophyll a (Chl a) and b (Chl b) levels, thereby improving photosynthesis [5,6,7]. Additionally, EBR can affect tomato photosynthesis by regulating stomatal size, increasing leaf area, and alleviating the suppression of tomato photosynthesis under high temperatures, thereby increasing carbon assimilation efficiency [8,9]. In rice, the mutation of the BR biosynthesis gene *Lhdd10* leads to suppressed chlorophyll synthesis, causing delayed heading and affecting rice yield [10]. Spraying wheat with HBL significantly increased its net photosynthetic rate [11]. Furthermore, BRs are capable of regulating the initial carboxylation activity of ribulose-1,5-bisphosphate carboxylase/oxygenase (Rubisco), thus impacting the assimilation of CO_2_ during photosynthesis [12,13,14]. BRs also enhance the light-capture efficiency of plants by inducing the transcription and translation of enzymes involved in chlorophyll biosynthesis [15]. In conclusion, BRs can maintain high chlorophyll content in plants and improve the net photosynthetic rate of leaves. It can also promote the transportation of photosynthetic products and reduce the inhibitory effect of stress on plant photosynthesis.

Previous research on photosynthesis has used genetic engineering to improve photosynthetic efficiency by enhancing light capture and energy conversion during the light reactions of photosynthesis, as well as carbon assimilation during dark reactions [16,17]. The thylakoid membranes of higher plant chloroplasts have four major protein complexes, arranged in a specific order, that work together to complete photosynthetic light reactions: photosystem II (PSII), cytochrome b6f, photosystem I (PSI), and ATP synthase [18]. PSI is associated with light-harvesting complex I (LHCI), and PSII is associated with light-harvesting complex II (LHCII), each of which forms complexes [18]. The LHCII consists of inner and outer light-harvesting antennas, wherein the inner antenna contains the CP43 and CP47 proteins, whereas the outer antenna contains six proteins: Lhcb1, Lhcb2, Lhcb3, Lhcb4 (CP29), Lhcb5 (CP26), and Lhcb6 (CP24) [19,20]. Lhcb4, Lhcb5, and Lhcb6, which are located between the outer antenna and the core complex, mainly exist as monomers, and they play crucial roles in maintaining the normal structure and function of PSII [21]. The most important function of the light-harvesting antenna protein complex (LHCII) is to absorb light energy from the environment and quickly transfer it to the photosynthetic reaction center to induce photochemical reactions, thereby converting light energy into chemical energy [22,23]. The efficiency of plant light capture can be improved by increasing the size of the light-harvesting antenna proteins in PSII-LHCII [18,24]. Antenna proteins such as Lhcb6 and Lhcb5 can improve the efficiency of light capture in plants, because these proteins are involved in the xanthophyll cycle and help mitigate light damage [25,26,27,28].

Decreases in the PSII quantum yield have been reported to occur in plants lacking Lhcb5 and Lhcb6. Specifically, the lack of Lhcb5 is linked to diminished PSII quantum yield due to the quenching of chlorophyll fluorescence at the Fm level. The absence of Lhcb5 or Lhcb6 does not lead to their replacement with another complex [29]. Arabidopsis lacking Lhcb6 will lead to a decrease in its photosynthetic rate and growth inhibition. The pigment and protein contents of the thylakoid membrane remain largely unchanged in other respects [29]. In addition, enhanced light-harvesting antenna proteins in PSII-LHCII can improve the light energy capture efficiency of plants [30]. However, the transcription factors regulating the light supplement antenna protein gene in maize remain to be further identified.

This research revealed that the transcriptome of maize seedlings treated with EBL highlights the influence of EBL on metabolic pathways associated with photosynthesis in maize leaves. EBL regulates genes involved in photosynthetic antenna proteins, as well as porphyrin and chlorophyll metabolism. Our findings indicate that EBL upregulates Lhcb5 and MYBR17, with co-expression analysis showing a moderate positive correlation between the transcription factor MYBR17 and Lhcb5. Additionally, MYBR17 enhances the activity of the Lhcb5 promoter. Further investigations identified MYBR17 transcription factor binding sites on the Lhcb5 promoter, located between −532 and −269 bp. These findings lay a theoretical foundation for utilizing EBL hormones to enhance maize yield.

## 2. Materials and Methods

### 2.1. Plant Material and Treatments

Maize seeds of the inbred line Mo17 were sown in soil and incubated at 28 °C with a light cycle 16 h of light followed by 8 h of darkness for approximately two weeks, until the seedlings developed three fully expanded leaves. These two-week-old plants were then utilized for the extraction of genomic DNA (gDNA) and total RNA, as well as for EBL hormone treatment [31]. The inbred line of maize Mo17 was cultivated in the experimental fields at Sichuan Agricultural University, Ya’an, China, following the prescribed agronomic practices, and underwent self-pollination. We gathered various tissues (including anther, silk, leaf, stem, root, 15 DAP seed, endosperm, and embryo) from the Mo17 inbred maize to analyze the specific expression patterns of MYBR17 and *Lhcb5* [32].

This study utilized seeds from the Arabidopsis Co-0 ecotype (wild-type, WT). The *AtMYBR17* (AT5G56840) T-DNA mutant lines (*mybr17*, SALK_200938) were sourced from The Arabidopsis Information Resource (TAIR). The seeds were surface sterilized and then stratified at 4 °C for 2 days prior to being sown on the medium. Both Col-0 and mutant plants were cultivated in growth chambers under a 16-h light/8-h dark cycle with 60% relative humidity at temperatures ranging from 18 to 22 °C, using either half-strength MS medium containing 0.9% agar or full-strength MS medium [33,34].

### 2.2. RNA Isolation, Gene Cloning, and Expression Analysis

Total RNA was extracted from the leaves of Arabidopsis and maize using a TRIzol reagent (Vazyme, Nanjing, China) following the manufacturer’s instructions. The cDNA was produced utilizing M-MLV reverse transcriptase (Takara, Beijing, China). The coding sequence (CDS) of MYBR17 was amplified from maize leaves and inserted into a pGEMT vector (Tiangen, Beijing, China) for sequencing. The confirmed MYBR17 gene was then subcloned into pBI221, driven by the maize ubiquitin promoter, for transient overexpression in maize protoplasts. Semi-quantitative reverse transcription PCR (RT-PCR) and quantitative real-time PCR (qRT-PCR) were conducted to assess gene expression levels. The qRT-PCR analyses were carried out on a CFX96 system (Bio-Rad, Hong Kong, China) using SsoFast EvaGreen Supermix (Bio-Rad). Relative gene expression was evaluated using the 2^−∆∆Ct^ method, with the maize *Actin* gene serving as the internal control [35,36]. All of the primers used for qPCR are listed in Appendix A.

### 2.3. Promoter Activation Assays

Promoter activation assays were conducted using the luciferase (LUC)–β-glucuronidase (GUS) reporter system. Fragments of the *Lhcb3 Lhcb5* and *Lhcb6* promoters (1.5–2 kb upstream of the start codon) were cloned from maize gDNA and inserted into the pBI221-Ubi-LUC vector, substituting the ubiquitin promoter [32,37]. The coding sequence (CDS) of MYBR17 was incorporated into pBI221-Ubi-GUS, replacing the *GUS* gene. The promoter–LUC constructs were co-transfected, alongside the MYBR17 overexpression construct pUbi-MYBR17, into maize leaf protoplasts as previously described. We used pBI221-Ubi-GUS as an internal control. Following 24 hours of incubation, the transfected protoplasts were placed in CCLR lysis solution (100 mM KH_2_PO_4_ (pH 7.8), 1 mM EDTA, 10% glycerol, 1% Triton X-100, and 7 mM β-mercaptoethanol) for protein extraction. GUS and LUC enzymatic assays were performed using 4-Methylumbelliferyl-β-D-glucuronide and luciferase assay reagent (Promega, Madison, WI, USA), respectively, as substrates [38]. Fluorescence and luminescence measurements were taken with a Fluoroskan Ascent FL Microplate Fluorometer and a Luminometer, respectively (Thermo, Rockford, IL, USA).

### 2.4. Analysis of MYBR17 Transactivation

The transactivation activity of MYBR17 was assessed using the pGBKT7 yeast expression plasmid, which is based on the GAL4 system [39]. The full-length coding sequence of MYBR17 was ligated in frame with the GAL4 binding domain at the *EcoR*I and *BamH*I sites of the pGBKT7 plasmid. The pGBKT7-GAL4 and pGBKT7 constructs served as the positive and negative controls, respectively. These constructs were introduced into the yeast strain AH109 using the Yeastmaker™ Yeast Transformation System, following to the manufacturer’s protocol. Yeast transformants were selected on SD/™Trp, and transactivation activity was evaluated on SD/™Trp/+X-α-gal plates. The yeast cells on the selection and screening plates were incubated at 28 °C for a duration of 3 to 4 days.

### 2.5. Subcellular Localization of MYBR17

To determine the subcellular localization, the CDS of MYBR17 was amplified and inserted into pCAMBIA2300 vector without including a stop codon, resulting in a MYBR17–enhanced green fluorescent protein (eGFP) fusion construct [38,40]. In summary, approximately 100 μL of protoplast cells at a density of 2 × 10^6^ cells/mL were transfected with 10 μg of plasmid DNA from the fusion construct and incubated at 28 °C for 16 hours. The intracellular signals of GFP and red fluorescent protein (RFP) in the protoplast cells were examined using a confocal microscope with the Leica Application Suite X (Leica Microsystems, Wetzlar, Germany).

### 2.6. Yeast One-Hybrid Analysis

In order to investigate the in vivo interaction of MYBR17 with the *Lhcb5* promoter, a yeast one-hybrid assay was conducted utilizing the Y187 yeast strain [31]. The coding sequence (CDS) for MYBR17 was subcloned into a pGADT7-Rec2 vector through *Nde*I and *BamH*I digestion. The bait sequence from the *Lhcb5* promoter fragment (−532 to −269 bp), which harbors the active MYB-binding site (−485 to −490 bp), was cloned and inserted into the yeast vector pHIS2 using *EcoR*I and *Sac*I. Following this, MYBR17 and its corresponding promoter constructs were co-transformed into the Y187 yeast strain. Positive clones were identified on a selective SD/−Trp/−His/medium supplemented with 50 mM Triazol-3-amine (3-AT) at 30 °C for 2 to 6 days. A complete list of primers utilized for qPCR is provided in Appendix A.

### 2.7. Analysis of Photosynthetic Parameters and Chlorophyll Content

Photosynthetic parameters were assessed using an Li-6400 XTR photosynthesis system (Li-COR; Lincoln, NE, USA). The light intensity was calibrated to 600 μmol·m^−2^·s^−1^, with the temperature maintained at 25 °C and CO_2_ concentration held at 400 μL·L^−1^. Light intensities were varied, set at levels of 0 μmol·m^−2^·s^−1^, 100 μmol·m^−2^·s^−1^, 200 μmol·m^−2^·s^−1^, 300 μmol·m^−2^·s^−1^, 400 μmol·m^−2^·s^−1^, 800 μmol·m^−2^·s^−1^, 1000 μmol·m^−2^·s^−1^, and 1200 μmol·m^−2^·s^−1^. Photosynthetic data were analyzed using dedicated software for simulating photosynthesis, while the contents of ALA and PBG were quantified following established methods [41,42]. The measurement of chlorophyll content was conducted as previously described [43].

### 2.8. Statistical Analysis

Student’s *t*-tests were conducted to analyze samples from three independent replicate experiments. Results are expressed as the mean ± standard error (SE) or mean ± standard deviation (SD). *p* < 0.05 was considered statistically significant (* *p* < 0.05, ** *p* < 0.01) using Student’s *t*-test.

## 3. Results

### 3.1. Differential Gene Expression Analysis of Photosynthetic Antenna Protein RNA-Seq

To determine whether the light-harvesting antenna proteins in maize leaves respond to EBR signaling, we performed transcriptome sequencing on maize leaves treated with 0.5 μM EBR for 24 h. We identified 2041 differentially expressed genes (DEGs) between the treatment and control groups, with 1051 upregulated and 990 downregulated DEGs. The DEGs were predominantly associated with five pathways: biosynthesis of secondary metabolites, ribosomes, porphyrin and chlorophyll metabolism, photosynthesis, and the photosynthetic antenna proteins (Appendix A). From the enriched pathways of the photosynthetic antenna protein DEGs, we preliminarily screened eight upregulated candidate genes (Table 1), including three photosynthetic antenna protein genes (*Lhcb3*, *Lhcb5*, and *Lhcb6*) and five transcription factor genes (*MYBR17*, *Sbp25*, *Jmj14*, *bHLH103*, and *bZIP76*).

To validate the RNA-Seq findings, real-time quantitative PCR (qPCR) was performed on the eight upregulated genes, yielding similar results. The results indicated that after the exogenous application of EBR, all eight genes were upregulated to varying degrees, with *Lhcb5* and the transcription factor *MYBR17* showing the most significant upregulation (Figure 1).

### 3.2. Identification of Candidate TFs That Bind to the Lhcb5 Promoter

To screen for transcription factors that may regulate *Lhcb3*, *Lhcb5*, and *Lhcb6*, we selected five upregulated transcription factors from transcriptome data. Based on the reference transcriptome data for leaves at different developmental stages provided by Stelpflug [44], we calculated the Pearson correlation coefficients (PCCs) (Appendix A). We validated the transcription factors bHLH103 and MYBR17, which had correlation coefficients greater than 0.5. To identify transcriptional activity in the promoter, we amplified *Lhcb3*, *Lhcb5*, and *Lhcb6* (1.5–2 kb upstream of the start codon). The promoters of *Lhcb3*, *Lhcb5*, and *Lhcb6* were inserted into the pBI221-Ubi-LUC vector to replace the ubiquitin promoter and control the LUC reporter gene. These promoter–LUC plasmids were introduced into maize protoplasts, with the LUC reporter gene under the control of the cauliflower mosaic virus (CaMV) *35S* promoter serving as an internal control. The LUC activities of *Lhcb3pro*, *Lhcb5pro*, *Lhcb6pro*, and pro35S were similar (Figure 2a). The reporter constructs, utilizing pUbi-MYBR17-GUS as the effector plasmid and pUbi-GUS as the internal control, were co-transfected into maize endosperm protoplasts for a transient expression assay. Following a 24 h incubation period, the activities of the GUS and LUC fluorogenic assays were analyzed.

Promoter–LUC plasmids were subsequently introduced into maize protoplasts either with or without the pUbi-MYBR17-GUS or pUbi-bHLH103-GUS expression constructs. As shown in Figure 2b, only co-transfection with MYBR17 markedly increased *Lhcb5pro*: LUC activity, indicating the activation of the *Lhcb5* promoter. In summary, these results demonstrate that the transcription factor MYBR17 specifically recognizes the *Lhcb5* promoter and induces transcriptional activation.

### 3.3. Analysis of the Expression Patterns of Candidate Transcription Factors and Target Genes

To investigate its subcellular localization, we attached the full-length MYBR17 to the C-terminus of an enhanced green fluorescent protein (eGFP), using free eGFP as a control. Both constructs were regulated by a constitutive *35S* promoter. These constructs were then transiently expressed in maize leaf mesophyll protoplasts. Analysis of subcellular localization revealed that MYBR17 was present in the nuclei of the maize protoplasts (Figure 3a).

Next, we conducted an analysis of the temporal and spatial expression pattern using reverse transcriptase quantitative PCR (RT-qPCR) across eight maize tissues (anther, silk, leaf, stem, root, seed, endosperm, and embryo) to validate the tissue-specific expression profiles of *Lhcb5* and *MYBR17*. The results from RT-qPCR indicated that the transcripts of *Lhcb5* and *MYBR17* have significant functions in maize leaves (Figure 3b). Additionally, the *Lhcb5* and *MYBR17* transcripts were moderately present in the seed and were also expressed to a lesser extent in the embryo, silk, root, and stem. Therefore, MYBR17 is likely to have regulatory functions in the development of maize endosperm and seed.

To investigate the autonomous transcriptional activity of MYBR17 in yeast, we ligated the full-length coding sequence of the MYBR17 in frame with the GAL4-BD of the pGBKT7 plasmid and subsequently transformed this construct (pGBKT7-MYBR17) into the yeast strain AH109. We also transformed pGBKT7-GAL4, known for its robust transactivation capability, and an empty pGBKT7 vector into the AH109 strain to serve as positive and negative controls, respectively. The pGBKT7-MYBR17 construct exhibited significant autonomous transcriptional activity in yeast, confirming its role as a positive control. (Figure 3c).

### 3.4. MYBR17 Activated the Promoters of Lhcb5

To clarify how MYBR17 interacts with and modulates the *Lhcb5* promoter, the Plant Care system was utilized to examine the cis-acting elements in the promoter region [45]. Five MYB-binding sites were distributed in the *Lhcb5* promoter at −485, −582, −885, +1290, and +1731 bp (Table 2). Therefore, we generated five fragments of promoter deletions of *Lhcb5* (Figure 4a). In order to further investigate the regulatory mechanism of MYBR17, we utilized a promoter–reporter system to assess the activation of Lhcb5 by analyzing five promoter fragments and performing a transient activity analysis to identify the region containing the core binding motif that MYBR17 targets. Notably, compared to the full-length promoter (*Lhcb5pro*-1674), the activities of promoter fragments of different lengths did not change significantly.

The promoter was constructed alongside pUbi-MYBR17-GUS as an effector plasmid, using pUbi-GUS as an internal control (Figure 4a), and both were transiently co-transfected into the maize leaf protoplasts for the expression assay. As shown in Figure 4b, the ratio of LUC/GUS activity in protoplasts transfected with *Lhcb5pro*-269: LUC + pUbi-MYBR17-GUS was markedly lower than that of protoplasts transfected with *Lhcb5pro*-532: LUC + pUbi-MYBR17-GUS. Notably, the activating effect of MYBR17 was entirely abolished in the cells transfected with *Lhcb5pro*-269: LUC + pUbi-MYBR17-GUS (Figure 4c), suggesting that the promoter segment between *Lhcb5pro* (−532 bp) and *Lhcb5pro* (−269 bp), approximately 263 bp, is critical for the transcription activation of *Lhcb5* by MYBR17.

Yeast one-hybrid assays were conducted to confirm the binding activity of MYBR17 with the *Lhcb5* promoter. The bait sequence of the *Lhcb5* promoter, a fragment ranging from −532 to −269 bp that includes the active MYB-binding site at −485 to −490 bp, was cloned into the yeast vector pHIS2. The CDS of *MYBR17* was subcloned into the vector pGADT7-Rec2. MYBR17 and its promoter constructs were co-transformed into the yeast strain Y187. The yeast cells exhibited normal growth in a selective medium containing 50 mM 3AT in the presence of MYBR17, indicating a direct interaction between MYBR17 and the Lhcb5 promoter fragments (263 bp) (Figure 4d). These findings confirm that MYBR17 directly binds to the MYB-binding site within the Lhcb5 promoter, thereby regulating gene expression.

### 3.5. MYBR17 Deficiency Reduces Photosynthesis in Arabidopsis

In order to further clarify the role of MYBR17, we created a T-DNA insertion mutant in Arabidopsis. This mutant was self-pollinated and cultivated for three generations to establish a homozygous T-DNA insertion line (Appendix A). Line SALK_200938 (named *mybr17*) was found to contain insertions with the coding region of *AtMYBR17* (Figure 5a). The homozygous *mybr17* mutant exhibited growth defects and significantly weaker leaf development than the WT plants. The leaves of mybr17 were significantly shorter and narrower, measuring 0.54 cm and 0.08 cm less in length and width, respectively, than those of the WT.

The *AtMYBR17* and *AtLhcb5* transcript levels in the mutant lines and WT plants were identified using quantitative real-time PCR (qPCR) analysis. The expression level of *AtLhcb5* was significantly reduced in the *mybr17* mutant. However, the *AtMYBR17* transcript was almost undetectable in the mutant due to the insertion of T-DNA into the coding region (Figure 5b).

We used a Li-6400 XTR photosynthesizer to quantify the net photosynthetic rate, transpiration rate, stomatal conductance, and intercellular carbon dioxide concentration in both WT and *mybr17* mutant leaves (Figure 5c,d). The mybr17 mutant leaves showed a notable decrease in net photosynthetic rate, transpiration rate, and stomatal conductance compared to the WT. Conversely, higher intercellular carbon dioxide concentration was observed in the leaves of the mutant than in those of the WT. Collectively, these results indicate that the reduced photosynthetic capability of the *mybr17* mutant leads to phenotypic changes.

In order to further elucidate the reduced photosynthetic ability of the *mybr17* mutant, we measured the chlorophyll content, PSII maximum photosynthetic rate (Fv/Fm), ΦPSII, NPQ, and qP in both the mutant and WT (Table 3). The total chlorophyll content in the *mybr17* mutant was significantly lower than that of the WT, but the ratio of chlorophyll a to b remained relatively unchanged. Additionally, the Fv/Fm, ΦPSII, NPQ, and qP values were all notably lower in the mutant compared to those of the WT. The decrease in PSII activity in these mutants was attributed to their reduced photosynthetic capacity.

## 4. Discussion

Plant hormones are indispensable bioactive substances that regulate various physiological and biochemical processes in the life cycle of plants. MYB transcription factors belong to one of the largest plant TF families and play a crucial role in plant hormone responses. There have been numerous reports on MYB transcription factors in response to plant hormones. In Arabidopsis, the expression of the *AtMYB2* gene is closely related to ABA signaling [46]. When exogenously active salicylic acid is applied, the expression of MYB transcription factors in tobacco increases [47]. Gibberellin signaling induces the expression of MYB genes in Arabidopsis, affecting flowering time and anther fertility [48]. Brassinosteroids (BRs) are the sixth major group of hormones involved in various plant processes. Numerous studies have demonstrated the regulatory roles of BRs in photosynthesis. Soybeans treated with BRs showed an increased maximum quantum yield of PSII and RuBisCO enzyme activity, leading to increased plant biomass accumulation and yield [49]. BRs also promote dark photosynthetic reactions. After the foliar application of brassinolide (BL) to wheat (*Triticum aestivum*) and mustard (*Brassica juncea*), brassinolide-regulated rice grain filling enzyme activity increased, and the accumulation of photosynthetic products in plants was stimulated [11]. Furthermore, the treatment of field maize leaves with BR increased the chlorophyll content and PEPC enzyme activity, significantly increasing the net photosynthetic rate by 32.6% [50,51]. Utilizing EBR to enhance the activity of transcription factors that regulate photosynthesis would be beneficial for improving maize yield. The regulation of photosynthesis by BRs involves multiple processes, such as stomatal opening and light and dark reactions. The regulation of photosynthesis by BRs has been found to be related to multiple genes. However, research on the molecular mechanisms by which BRs affect the expression of *Lhcb5* is still relatively scarce [44]. In this study, after spraying maize seedling leaves with 0.5 μmol/L EBR for 24 h, we found that a large number of transcription factors and functional genes involved in the regulation of photosynthesis were induced to be upregulated (Table 1), which again demonstrates the positive regulation of photosynthesis by BRs. The transcription factor MYBR17 showed a notable increase in expression, particularly in the leaves, suggesting it may play a role in regulating photosynthesis.

The MYB family of transcription factors plays an active role in the regulation of plant photosynthesis. In Arabidopsis, the MYB-related transcription factors AtCCA1 (Circadian Clock Associated 1) can bind to a region of the promoter of an Arabidopsis light-harvesting chlorophyll a/b protein gene, *Lhcb1*3*, which is necessary for its regulation by phytochrome [52,53,54]. HvMCB1 and HvMCB2 (*Hordeum vulgare*) and other unknown regulators may mediate the transcription of the *CAB* gene (encoding the chlorophyll a/b-binding protein of photosystem II), enhancing the plant’s responsiveness to various environmental and developmental signals [55]. PnMYB2, an MYB transcription factor in *Panax notoginseng*, participates in photo cooperation through jasmonic acid signaling to regulate the resistance of *Panax notoginseng* pathogens [56]. The R2R3-MYB transcription factor BpMYB106 enhances photosynthesis and growth rate by upregulating the expression of photosynthetic genes [57]. Furthermore, under salt stress, the MYB family of transcription factors is also involved in the regulation of plant photosystems I and II [58]. In this study, MYBR17 was found to be significantly induced by BRs in maize. In addition, the results of dual-luciferase reporter assays and yeast one-hybrid experiments showed that MYBR17 can bind to the promoter region of the light-harvesting antenna protein *Lhcb5* and significantly upregulate its expression.

MYB transcription factors have been found in many higher plants. Researchers have analyzed their homology through the evolutionary tree of the MYB family in different plants, and have found that Arabidopsis is a feasible model plant for the study of the regulatory mechanisms of the transcription factor MYB [59,60]. In maize, we previously discovered that MYBR17 binds to the promoter region of the light-harvesting antenna protein *Lhcb5* and significantly upregulates its expression. Further studies on Arabidopsis T-DNA insertional mutants found that, compared to WT Arabidopsis, the length and width of mutant leaves were reduced to varying degrees, the levels of *AtLhcb5* expression and total chlorophyll content were significantly decreased, and the photosynthetic parameters of the mutant Arabidopsis showed a significant downward trend compared to the WT. These results indicate that the absence of the MYBR17 transcription factor reduces the expression of the light-harvesting antenna protein *Lhcb5*, leading to a decrease in the photosynthetic rate of the leaves, providing strong support for the study of plants overexpressing MYBR17 to improve the photosynthetic rate, providing an evidentiary foundation for an increase in the yield of maize.

## 5. Conclusions

This research revealed that the transcriptome of maize seedlings treated with EBL highlights the influence of EBL on metabolic pathways associated with photosynthesis in maize leaves. EBL regulates genes involved in photosynthetic antenna proteins, as well as porphyrin and chlorophyll metabolism. Our findings indicate that EBL upregulates *Lhcb5* and MYBR17, with co-expression analysis showing a moderate positive correlation between the transcription factor MYBR17 and *Lhcb5*. Additionally, MYBR17 enhances the activity of the *Lhcb5* promoter. Further investigations identified MYBR17 transcription factor binding sites on the *Lhcb5* promoter, located between −532 and −269 bp. These findings lay a theoretical foundation for utilizing EBL hormones to enhance maize yield.

## Figures and Tables

**Figure 1 biomolecules-15-00094-f001:**
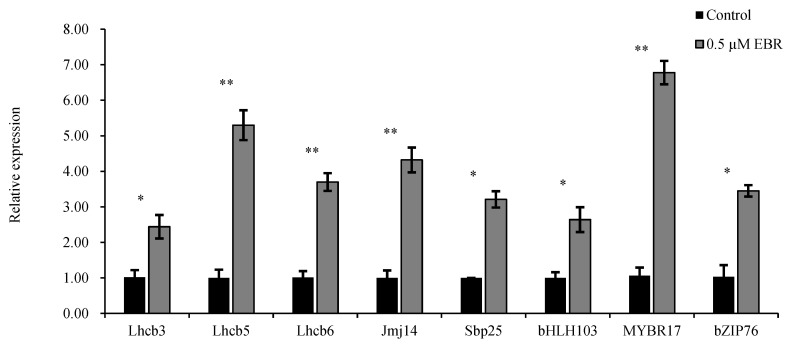
Gene expression analysis of candidate transcription factors and related target genes. * *p* < 0.01, ** *p* < 0.001.

**Figure 2 biomolecules-15-00094-f002:**
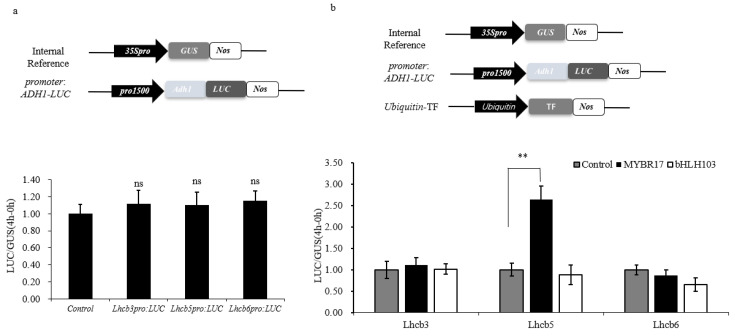
Promoter activity analysis and candidate transcription factor identification. (**a**) Analysis of promoter activity of *Lhcb3pro Lhcb5pro* and *Lhcb6pro*. (**b**) Candidate transcription factors regulate target gene promoters. (**Top**) The schematic diagrams of the reporter constructs and (**bottom**) the dual-luciferase reporter (DLR) assay. ** *p* < 0.001; ns, no significant difference.

**Figure 3 biomolecules-15-00094-f003:**
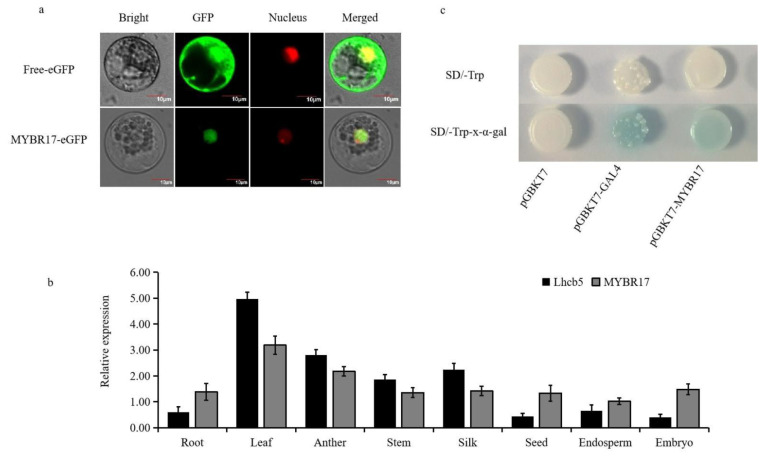
Spatiotemporal expression pattern and transcription factor characteristics of MYBR17. (**a**) Subcellular localization of MYBR17-eGFP in maize protoplasts. Bars = 10 μm. (**b**,**c**) Analysis of MYBR17 transactivation in the yeast system, with pGBKT7-GAL4 and pGBKT7 as the positive and negative controls, respectively.

**Figure 4 biomolecules-15-00094-f004:**
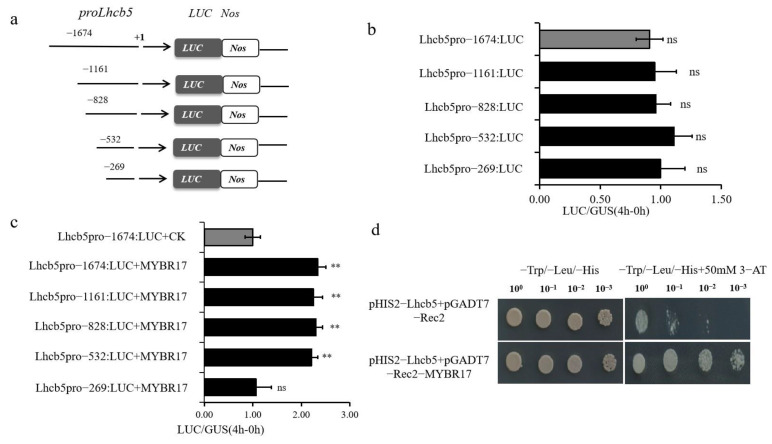
Preliminary study on the binding region of transcription factors MYBR17 to *Lhcb5* promoter. (**a**) *Lhcb5* promoter fragment deletion vector construction diagram. (**b**) Analysis of interaction region between *Lhcb5* promoter and MYBR17. (**c**) Analysis of interaction region between *Lhcb5* promoter. (**d**) Yeast one-hybrid analysis. ** *p* < 0.01; ns, no significant difference.

**Figure 5 biomolecules-15-00094-f005:**
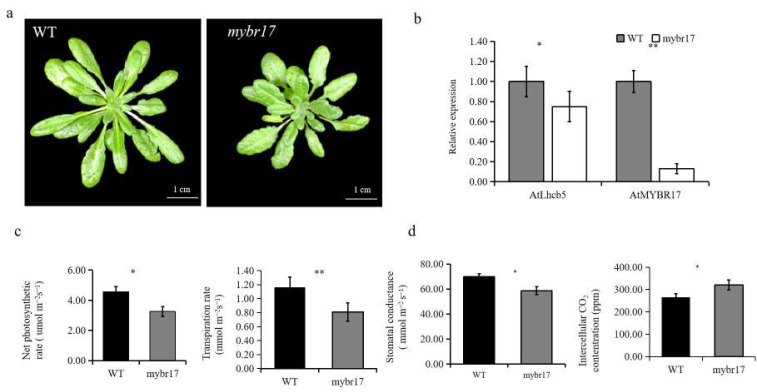
Analysis of Arabidopsis *mybr17* mutants. (**a**) Arabidopsis wild-type and *mybr17* plants. Bars = 1 cm. (**b**) RT-qPCR analysis of *AtMYBR17* and *AtLhcb*5 expression in the leaf of *mybr17* mutant. (**c**,**d**) The net photosynthetic rate, transpiration rate, stomatal conductance, and intercellular carbon dioxide concentration of both WT and *mybr17* mutant leaves. * *p* < 0.05, ** *p* < 0.01.

**Table 1 biomolecules-15-00094-t001:** Genes preliminarily screened by transcriptome.

Gene ID	Gene Name	Down- or Upregulated	log2 Fold-Change
Zm00001d009589	Lhcb3	Up	0.903876796
Zm00001d007267	Lhcb5	Up	1.437698745
Zm00001d433540	Lhcb6	Up	0.696991975
Zm00001d033957	bHLH103	Up	1.007641981
Zm00001d014698	Sbp25	Up	0.839243932
Zm00001d044409	MYBR17	Up	1.706390574
Zm00001d036736	bZIP76	Up	1.020292463
Zm00001d030108	JMJ14	Up	1.109304348

**Table 2 biomolecules-15-00094-t002:** Analysis of cis-acting elements in *Lhcb5pro*.

Cis-Element	Position	Signal Sequence	Putative Function
CAAT-box	CAAT/CCAAT/CAAAT	+146, −256, +495, +856, −125, +1800	Common cis-acting element in promoter and enhancer regions
MYB	CAACCA/TAACCA	−485, −582, −885, +1290	Myb-binding site
BRRE	CGTGCG	+445, +846	Brassinolide responsive element
GARE-motif	TCTGTTG	+1820	Gibberellin-responsive element

**Table 3 biomolecules-15-00094-t003:** Determination of photosynthetic parameters of WT and *mybr17*.

Parameter	WT	mybr17
Total chlorophyll (mg/g FW)	2.11	1.52
Chlorophyll a/b	3.59	3.56
Fv/Fm	0.81	0.75
ΦPSII	0.28	0.25
NPQ	1.13	0.95
qP	0.52	0.47

## Data Availability

No new data were created.

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
