# Peer review of "Epibrassinolide Regulates Lhcb5 Expression Though the Transcription Factor of MYBR17 in Maize"

_biomolecules, 2025, doi:10.3390/biom15010094_

Round 1
Reviewer 1 Report
Comments and Suggestions for Authors
The authors conducted transcriptome analysis followed by genetic studies of MYBR17 and Lhcb5 to show that MYBR17 directly regulates Lhcb5 and could be used to improve photosynthetic efficiency. The work also establishes a relationship between BR and maize leaf photosynthesis.
A primary concern is that although the authors conducted genetic analysis in maize background, transgenic analysis was conducted in Arabidopsis. The authors need to provide a reason for the inconsistency. Have you considered the MaizeGDB mutant lines? Have you checked the expression of these two genes in Arabidopsis? What could be the phenotype/photosynthetic rate of overexpression of MYBR17 in maize?
The authors may revise the title to include the plant species (maize).
What was the screening strategy used to filter 8 genes from DEGs? Are the criteria based on the fold change or other factors? Were there any downregulated genes in the DEG list related to photosynthesis? The authors may share the details of DEG analysis and raw data accession ID in the supplementary files.
Line 209-214: Details about the methods need to be moved to the materials and method section
The order of Figures 3b and 3c can be exchanged for consistency with their citation order in the manuscript.
Author Response
Comments 1:A primary concern is that although the authors conducted genetic analysis in maize background, transgenic analysis was conducted in Arabidopsis. The authors need to provide a reason for the inconsistency. Have you considered the MaizeGDB mutant lines? Have you checked the expression of these two genes in Arabidopsis? What could be the phenotype/photosynthetic rate of overexpression of MYBR17 in maize?
Response 1:Thank you for pointing this out. Due to the lack of suitable mutants for this gene in the maize mutant library, only the model plant Arabidopsis mutants can be used for functional verification. We have examined the expression of these two genes in Arabidopsis thaliana and the results are shown in Figure 5. We don't really know the phenotype/photosynthetic rate of overexpression of MYBR17 in maize cause we don't have the overexpression of MYBR17 material.
Comments 2:The authors may revise the title to include the plant species (maize).
Response 2:Thank you for pointing this out. We have added the plant species in title.
Comments 3:Line 209-214: Details about the methods need to be moved to the materials and method section
Response 3:Thank you for pointing this out. We have move the details into the materials.
Comments 4:The order of Figures 3b and 3c can be exchanged for consistency with their citation order in the manuscript.
Response 4:Thank you for pointing this out. We have change the Figures 3b and 3c order in the manuscript..
Reviewer 2 Report
Comments and Suggestions for Authors
The authors in this manuscript describe the molecular interaction between MYBR17 transcription factor and Lhcb5 gene of the light harvesting antenna in maize. The transcription factor which is itself brassinosteroid inducible, binds the promoter of Lhcb5 to activate its expression. The authors used RNA-seq data of BR-treated vs untreated maize plants to identify induced genes and validated the expression via RT-qPCR. Then used luciferase-based approaches and yeast one hybrid assays to show that MYBR17 binds to specific sites in the promoter of Lhcb5. Then to validate this finding they used a forward genetic approach, showing that mybr217 T-DNA mutant in Arabisdopsis thaliana has reduced Lhcb5 expression and somewhat impaired photosynthesis.
The findings in this manuscript are interesting and valiable to the scientific community.
Some minor comments
-the title begins with EBL which is confusing to people that do not know this abbreviation, authors should use the whole name of the hormone, also it would be useful to specify the plant they are using
-line 244 the experimental evidence does not support this claim, most tissues express both genes, the higher expression in the leaves suggests that the genes have a major function there but other claims are not well supported
-in Fig3.b in the negative control pGBKT7 the yeast is growing even in the -His media, i don't understand how this is possible since the pGBKT7 plasmid has only the TRP selection gene, how is the HIS reporter activated?please explain this result
-for Fig2, it would be interesting to show the sequence of the promoters of all Lhcb genes aligned and show potentially why only Lhcb5 is activated by MYBR17 and not the others
-the experiments with the Arabidopsis mutant are interesting but some supporting information are needed especially since only one mutant is shown
a model of the gene with the site of the T-DNA insertion could be beneficial for the readers, potentially also a model of the protein with the predicted domains
the authors should include a molecular characterization of the mutations, does it produce full length transcript but with reduced expression or the transcript is incomplete, is the mutant knock-out or knock down?
-it would be interesting to see what happens of you treat Arabidopsis plants with brassinolide, are both MYBR17 and Lhcb5 induced? is Lhcb5 induced in the mybr17 mutant?
these experiments would provide some validation for the model the authors propose
Closing, please provide a list of the accession numbers of the Arabidopsis and maize genes if possible to avoid potential confusions, since MYBs are a very large family with similar sequences in many cases.
Comments on the Quality of English LanguageEnglish need some improving but they do not affect the understanding of the manuscript
Author Response
Comments 1:the title begins with EBL which is confusing to people that do not know this abbreviation, authors should use the whole name of the hormone, also it would be useful to specify the plant they are using.
Response 1:Thank you for pointing this out. We have change the EBL to epibrassinolide in title.
Comments 2:line 244 the experimental evidence does not support this claim, most tissues express both genes, the higher expression in the leaves suggests that the genes have a major function there but other claims are not well supported
Response 2:Thank you for pointing this out. We have change the word in line 256-261 that the transcrips of Lhcb5 and MYBR17 were have significant functions in maize leaves and MYBR17 is likely to have regulatory functions in the development of maize endosperm and seed.
Comments 3:in Fig3.b in the negative control pGBKT7 the yeast is growing even in the -His media, i don't understand how this is possible since the pGBKT7 plasmid has only the TRP selection gene, how is the HIS reporter activated?please explain this result
Response 3:Thank you for pointing this out. We should be using the -Trp supplemented with X-α-gal as screening plate. it is clerical error.
Comments 4:for Fig2, it would be interesting to show the sequence of the promoters of all Lhcb genes aligned and show potentially why only Lhcb5 is activated by MYBR17 and not the others
Response 4:Thank you for pointing this out. Each Lhcb genes promoter is not same, The experiment results show that MYBR17 could regulate the Lhcb5 promoter active.
Comments 5:the experiments with the Arabidopsis mutant are interesting but some supporting information are needed especially since only one mutant is shown. a model of the gene with the site of the T-DNA insertion could be beneficial for the readers, potentially also a model of the protein with the predicted domains. the authors should include a molecular characterization of the mutations, does it produce full length transcript but with reduced expression or the transcript is incomplete, is the mutant knock-out or knock down?
Response 5:Thank you for pointing this out. As shown in fig 5, the mutant is knock down and T-DNA is inserted into the gene domain.
Comments 6:it would be interesting to see what happens of you treat Arabidopsis plants with brassinolide, are both MYBR17 and Lhcb5 induced? is Lhcb5 induced in the mybr17 mutant?
these experiments would provide some validation for the model the authors propose
Closing, please provide a list of the accession numbers of the Arabidopsis and maize genes if possible to avoid potential confusions, since MYBs are a very large family with similar sequences in many cases.
Response 6:Thank you for pointing this out. We will carry out relevant experiments in the future. The maize gene ID is GRMZM2G071977 and the Arabidopsis ID is AT5G56840.
Reviewer 3 Report
Comments and Suggestions for Authors
this work about EBL regulates Lhcb5 expression though the transcription factor of MYBR17. the results found that analyzing the transcriptomic data 19 of maize seedling leaves treated with 24-epibrassinolide (EBR) and verifying them using qPCR ex- 20 periments, we found that the MYBR transcription factor may regulate the expression of the pho tosynthetic light-harvesting antenna protein gene. Further experiments using protoplast transient expression and yeast one-hybrid tests showed that the maize transcription factor MYBR17 responds 23 to EBR signals and binds to the promoter of the light-harvesting antenna protein Lhcb5, thereby upregulating its expression. These results were validated using an Arabidopsis mybr17 mutant. Our findings provide a theoretical basis for the application of BRs in improving maize photosynthetic efficiency..
the idea is good but i still have some comments
1- the abstract is too short
2-=introduction
line 33 i suggest you to read this
5-Aminolevulinic acid and 24-epibrassinolide improve the drought stress resilience and productivity of banana plants
plz added paragraph about photosynthesis and their relation with your work
the aim of your study is not clear
your figs quality is bad even the asters away from the colum
check your statstic
'discussion part is weak
Comments on the Quality of English Language
need to carefully check
Author Response
Comments 1:the abstract is too short
Response 1:Thank you for pointing this out. Due to the magazine's request the abstracts not to exceed 200 and our abstract is 184.
Comments 2:introduction
line 33 i suggest you to read this
5-Aminolevulinic acid and 24-epibrassinolide improve the drought stress resilience and productivity of banana plants
plz added paragraph about photosynthesis and their relation with your work
the aim of your study is not clear
your figs quality is bad even the asters away from the colum
check your statstic
'discussion part is weak
Response 2:Thank you for pointing this out. I am so sorry that we did not find that sentence om line 33. Meanwhile many changes have been made to the article.
Round 2
Reviewer 2 Report
Comments and Suggestions for Authors
In this revised manuscript the authors address a lot of the points the reviewers suggested. The language has been improved and more details have been provided about the material and methods. The authors did not performed additional experiment with Arabidopsis suggesting that will be addressed in future work and since the focus of the paper is maize their point is valid.
In all the manuscript has been significantly improved in its current form. The only minor problems i found with this revised manuscript is the quality of photo in figure 4 which could be improved and that the supplemented material were not available.
Author Response
Comments 1:In all the manuscript has been significantly improved in its current form. The only minor problems i found with this revised manuscript is the quality of photo in figure 4 which could be improved and that the supplemented material were not available.
Response 1:Thank you for pointing this out. We have modified the format of Fig 4 and resubmitted supplementary materials.
Reviewer 3 Report
Comments and Suggestions for Authors
the authors did most of corrections that i asked
Comments on the Quality of English Languageminor revission
Author Response
Comments 1:the authors did most of corrections that i asked
Response 1:Thank you for pointing this out. We have made modifications to your problem, but only one question that we can't find the bananas you said. I also searched bananas and found no relevant references. Or maybe we didn't notice.